# Glycerol-Mediated Facile Synthesis of Colored Titania Nanoparticles for Visible Light Photodegradation of Phenolic Compounds

**DOI:** 10.3390/nano9111586

**Published:** 2019-11-08

**Authors:** Rab Nawaz, Chong Fai Kait, Ho Yeek Chia, Mohamed Hasnain Isa, Lim Wen Huei

**Affiliations:** 1Fundamental and Applied Sciences Department, Universiti Teknologi PETRONAS, Seri Iskandar 32610, Perak, Malaysia; 2Civil and Environmental Engineering Department, Universiti Teknologi PETRONAS, Seri Iskandar 32610, Perak, Malaysia; yeekchia.ho@utp.edu.my; 3Civil Engineering programme, Faculty of Engineering, Universiti Teknologi Brunei, Tungku Highway, Gadong BE1410, Brunei Darussalam; hasnain_isa@yahoo.co.uk; 4Advanced Oleochemical Technology Division (AOTD), Malaysian Palm Oil Board (MPOB), Bandar Baru Bangi, Kajang 43000, Selangor, Malaysia; limwen@mpob.gov.my

**Keywords:** colored titania, glycerol, Ti^3+^ ions, visible light absorption, photocatalytic performance, phenolic compounds

## Abstract

In this study, we developed a glycerol-mediated safe and facile method to synthesize colored titania nanoparticles (NPs) via solution route. Our method is considerably effective and greener than other options currently available. Colored titania NPs were produced by hydrolyzing TiCl_4_ precursor in aqueous solution containing different concentrations of glycerol (0.0, 1.163, 3.834, and 5.815 mol/L) and subsequent calcination at 300 °C for 1 h. Our results highlight firstly that glycerol-mediated synthesis is unlikely to affect the anatase crystalline structure of TiO_2_, and secondly, that it would lead to coloration, band gap narrowing, and a remarkable bathochromic redshift of the optical response of titania. More importantly, the synthesized colored titania have Ti^3+^ ions, which, at least in terms of our samples, is the major factor responsible for its coloration. These Ti^3+^ species could induce mid gap states in the band gap, which significantly improve the visible light absorption capability and photocatalytic performance of the colored titania. The photocatalytic experiments showed that the colored TiO_2_ NPs prepared in 1.163 mol/L aqueous glycerol solution displayed the best photocatalytic performance. Almost 48.17% of phenolic compounds and 62.18% of color were removed from treated palm oil mill effluent (POME) within 180 min of visible light irradiation.

## 1. Introduction

TiO_2_ is a semiconductor metal oxide that is used in a variety of applications, including, but not limited to, photocatalytic hydrogen generation [1], organic synthesis of chemicals [2], dye-sensitized solar cells [3], and environmental remediation [4,5,6]. When TiO_2_ is used as a photocatalyst, it produces excited charges (electron and hole) upon irradiation by light energy higher than its bandgap. These charge carriers transfer to the TiO_2_ surface to carry out the photocatalytic reaction, before their recombination [7].

The overall performance and photoactivity of TiO_2_ can be strongly affected by the amount and duration of excited charges on its surface. The number of charge carriers is dependent on light absorption. The more light absorbed by TiO_2_, the more likely the working electron and hole are present on the surface to perform photocatalytic reactions [8]. Because of the wide bandgap of TiO_2_ (3.0–3.2 eV), it is unable to efficiently absorb visible light, which makes up 45% of the solar radiations [1,9]. Thus, it has become a common research hotspot to shift the optical response of TiO_2_ into the visible light range and narrow its band gap in order to improve its overall efficiency for solar-driven visible light photocatalysis.

Recently, the discovery of hydrogenated black TiO_2_ in 2011 has prompted great research interest in the preparation of black TiO_2_ nanomaterials due to its efficient and improved visible light absorption capability and remarkable photocatalytic properties [10]. Since then, researchers have brought about some structural and chemical modifications which can turn white TiO_2_ into various colors like yellow, blue, grey, and black TiO_2_ using different starting materials and experimental conditions [11,12,13,14]. Certain chemical and structural modifications, including the creation of surface oxygen vacancy (SOVs) or Ti^3+^ species [15], formation of surface disorder core–shell structure [16], hydroxyl groups, and Ti–H bonding, have been made to produce colored TiO_2_. These modifications are believed to significantly alter the electronic structure and reduce the bandgap of TiO_2_, which results in improved visible light absorption. Moreover, due to these modifications, the adsorption and dissociation of reactants on the surface and the concomitant excited charges transport also improve, resulting in enhanced photocatalytic activity of colored TiO_2_ [17].

A range of reduction approaches has been employed to successfully synthesize colored TiO_2_, especially black TiO_2_ [17]. Most of the reduction approaches utilize hydrogen gas (hydrogenation) at high pressure [18], in some cases mixed with nitrogen and argon [19]. However, it is somehow risky, complicated, and cumbersome to prepare black TiO_2_ by using hydrogen gas directly because of its high flammability. Very few research groups have investigated reduction methods using solution routes or hydrogen sources other than pure H_2_ gas. For instance, Zou et al. synthesized black TiO_2_ using ascorbic acid and imidazole as reductants [20]. They reported that imidazole could effectively create SOVs and Ti^3+^ ions in the TiO_2_ matrix, resulting in reduced TiO_2_. In another work, they prepared grey TiO_2_ by mixing amorphous white TiO_2_ powder with hydrochloric acid (HCl) and imidazole, followed by calcination at 450 °C in a preheated furnace for 6 h [21]. Zuo and co-workers reported another synthetic method where they mixed ethanol, HCl, 2-ethyl imidazole, and titanium(IV) isopropoxide, and prepared reduced TiO_2_ (TiO_2−*x*_) by combustion of this mixture at 500 °C for 5 h [21]. Chen et al. synthesized black TiO_2_ by calcining a suspension of pristine TiO_2_ and ethanol in a vacuum furnace at 400 °C for 3 h [22]. Yin et al. prepared grey TiO_2_ nanowires via aluminum reduction at 700 °C for 4 h [23]. Nevertheless, besides its toxicity, the solvents and reductants used in these methods are neither cheaper nor easy to handle. Moreover, the Ti^3+^ ions and SOVs created by the existing methods are usually unstable because the reduction process occurs primarily on the surface of TiO_2_. Therefore, the quest for a promising, cost-effective, safe, and facile method for synthesis of colored TiO_2_ is in demand and still in progress.

During the last decade, glycerol, a byproduct of biodiesel, has attracted worldwide research attention in organic synthesis due to its distinctive physicochemical properties, such as low toxicity, high polarity, biodegradability, and high boiling point. Furthermore, glycerol has the ability to form robust hydrogen bonding, and dissolve both inorganic and organic salts [24]. It has been successfully utilized as a solvent, a co-solvent with ethanol, water, and any other polar solvent for the preparation of a variety of nanomaterials [24,25]. Recent work has revealed that glycerol can act both as a solvent and a reducing agent [26]. Considering the environmental legislation and replacement of the most frequently employed hydrogen sources, the use of glycerol is not only economically appealing but also a more promising alternative to develop environmentally friendly and sustainable processes for material synthesis [27,28]. The use of glycerol as solvent, co-solvent, and reductant is a fascinating area of research and gaining tremendous interest in the development of a sustainable and greener protocol for nanomaterials synthesis.

Motivated by the aforementioned considerations, the major objective of the study was to develop a green and facile method for synthesizing colored TiO_2_ nanoparticles (NPs). In order to achieve this objective, some specific research tasks were carried out. The tasks involved investigating for the first time the utilization of glycerol, a by-product from biodiesel production, as a green solvent to produce colored TiO_2_ NPs, to characterize synthesized colored TiO_2_ NPs using various techniques, and to assess the effectiveness and photocatalytic performance of colored TiO_2_ NPs for the decomposition of phenolic compounds and color removal from treated palm oil mill effluent (POME) under visible light irradiation. The colored TiO_2_ NPs were synthesized by hydrolyzing TiCl_4_ in an aqueous glycerol solution, followed by calcination at 300 °C for 1 h. Various colored TiO_2_ NPs, like off-white to light grey, grey, and black, were synthesized depending on the concentration of glycerol. The results reveal that the synthesized colored TiO_2_ NPs improved visible and infrared light absorption, narrow bandgap, and Ti^3+^ ions. The photodegradation results show that 48.17% photocatalytic degradation of the total phenolic compounds (initial concentration, 224.85 mg/L as gallic acid equivalent, GAE) and 62.18% of the color removal (initial concentration, 2420 (PtCo)) were obtained in 180 min by colored TiO_2_ NPs prepared in 1.163 mol/L aqueous glycerol solution and calcined at 300 °C for 1 h.

## 2. Experimental

### 2.1. Materials

Titanium tetrachloride (TiCl_4_, 99.9%), glycerol (85%), ammonium hydroxide (NH_3_ eq. 30%), Folin–Ciocalteau phenol reagent (99%), sodium carbonate (Na_2_CO_3_, 99.9%), and gallic acid were purchased from Merck (Darmstadt, Germany) and were used as received. Treated POME sample was obtained from a palm oil mill in Perak, Malaysia. The sample was collected in a 5 L plastic bottle and covered with a black plastic sheet to avoid light exposure during transportation, and stored at a temperature of 4 °C. The sample was filtered using Whatman^®^ membrane filter (pore size, 0.45 µm, Maidstone, UK) to remove the residual solids prior to analysis.

### 2.2. Synthesis of Colored TiO_2_ Nanoparticles

Colored TiO_2_ NPs were synthesized via hydrolysis of TiCl_4_ in aqueous glycerol solution, followed by calcination. In a typical synthesis procedure, approximately 0.1 mol of TiCl_4_ was transferred dropwise to 50 mL of aqueous glycerol solution. The glycerol and water were mixed in different ratios (*v*/*v*) to make the different concentrations of glycerol, as given in Table 1. The precipitation process was conducted by adding approximately 300 mL of 2.5 M NH_3_ (aq.) under vigorous stirring until pH 10 to get precipitate which was rinsed several times in deionized water. Lastly, the synthesized material was calcined at 300 °C for 1 h in a muffle furnace (Nabertherm, Lilienthal, Germany).

### 2.3. Materials Characterization

Raman spectra analysis was carried out to determine the crystalline structure of the synthesized material using Inspector 500 with an excitation wavelength of 532 nm. Diffuse reflectance (DR) UV-Vis spectra were acquired using Agilent Carry 100 (Santa Clara, CA, USA) with Spectralon as the reference material to determine the absorption edge of the as-synthesized material. The nitrogen (N_2_) adsorption–desorption isotherms of the material were acquired at −196 °C using the Micromeritics ASAP 2020 instrument (Micromeritcs Corps., Norcross, GA, USA). The samples were degassed at 300 °C for 12 h prior to measurements. The Brauner-Emmet-Teller (BET) method was used to calculate the specific surface area of the samples from adsorption data. The Barret–Joye–Halenda (BJH) method was followed for the pore size distribution analysis using desorption measurements. X-ray photoelectron spectroscopy analysis was carried out by Thermo Scientific spectrometer (K-alpha, Madison, WI, USA). Al Kα was used as an X-ray excitation source with C1s correction at 284.6 eV for calibration. Field emission scanning electron microscopy (FESEM) micrographs of the material were captured at 100 kX magnification using a Carl Zeiss (SUPRA 55VP, Oberkochen, Germany) instrument which was operated at 50 kV acceleration voltage.

### 2.4. Analytical Methods

The color concentration of the treated POME sample before and after the reaction was determined using a closed reflux colorimetric technique with help of DR3900 HACH spectrometer (Berlin, Germany) [29,30]. The color concentration was measured at 455 nm using platinum–cobalt (PtCo) as standard. Total phenolic compounds concentration was examined spectrophotometrically through a slight modification of the Folin–Ciocalteau (F-C) analytical technique, as described by Ergul et al. [31]. The phenolic compounds were analyzed using gallic acid as the reference standard compound [32] and the concentration of phenolic compounds was taken into consideration as total phenolic compounds (TPC) and expressed as mg/L of gallic acid equivalent (GAE). No quantitative analysis of individual phenolic compounds and their intermediates was taken into account. The F-C reagent contains phosphomolybdic and phosphotungstic complexes. This relies on the transfer of electron in alkaline medium from phenolic compounds to form a blue chromophore constituted by phosphomolybdic and phosphotungstic complexes, where the maximum absorbance depends on the concentration of phenolic compounds.

In a typical procedure, 0.1 mL of five-fold diluted treated POME sample was added to 0.5 mL of four-fold diluted F-C phenol reagent. After 5 min of retention time, the sample was kept for an hour at ambient temperature after the addition of 0.5 mL of Na_2_CO_3_ (200 g/L). Finally, the absorbance measurement was recorded at 765 nm with the help of a spectrophotometer, SpectroVis Plus instrument (Beaverton, OR, USA) and compared with the calibration curve of gallic acid standards.

### 2.5. Photocatalytic Experiments

Laboratory scale experiments were performed to assess the photocatalytic performance of the colored TiO_2_ NPs by monitoring the phenolic compound degradation and color removal from treated POME. All the photocatalytic reactions were performed in a 250-mL Pyrex photoreactor (7.0 cm ID × 9.0 cm height) with a quartz window. The window was tightly sealed at the top of the photoreactor to avoid evaporation of the working solution. The lamp was positioned 10 cm above the photoreactor and cooled by an electric cooling fan to maintain the reaction temperature at ~30 °C. The schematic diagram of the experimental setup for carrying out photocatalytic reactions is presented in Figure 1.

The colored titania sample (0.8 g/L) was suspended in 50 mL of treated POME solution (concentration of total phenolic compounds was 224.85 mg/L GAE and color 2420 PtCo). In order to achieve adsorption–desorption equilibrium between the photocatalyst and the phenolic compounds, the suspension was vigorously stirred in the dark for 30 min. The suspension was irradiated with visible light from a 500 W halogen lamp for 180 min. Controlled experiments under the same conditions without the presence of a catalyst or direct photolysis were carried out for comparison purposes. Vigorous stirring was sustained throughout the reactions to keep TiO_2_ NPs suspended in the treated POME solution. At a specific time interval (30, 60, 120, and 180 min), an aliquot of 1 mL was withdrawn from the photoreactor using a high precision syringe. Each sample was filtered using a Captiva Econo Filter (Nylon, diameter 0.2 µ, Santa Clara, CA. USA) prior to analysis. The concentration of total phenolic compounds as GAE was determined from the peak intensity of GA measured at 765 nm and color concentration (PtCo) at 455 nm with a spectrophotometer. The degradation efficiency of phenolic compounds and color removal were calculated using Equation (1):(1)$$X\;\%=\;\frac{C_{i}-C_{t}}{C_{i}}\;\times 100\%,$$
where X represents the percentage of degradation of phenolic compounds and color removal, and C_i_ and C_t_ denote their initial and concentrations at time (t), respectively.

To evaluate the stability of the colored TiO_2_ NPs, the spent photocatalyst was recycled to test its reusability. The recycled photocatalyst was used for three consecutive cycles after thoroughly washing it with deionized water before each run.

## 3. Results and Discussion

### 3.1. Synthesis Process of Colored TiO_2_

In the current work, we developed a cost-effective, safe, facile, and green synthesis route to prepare colored TiO_2_ NPs with enhanced visible light absorption. TiCl_4_ was used as the precursor and glycerol–water as co-solvents. A viscous solution was obtained by the dropwise addition of TiCl_4_ into a reaction media containing 0.0, 1.163, 3.834, and 5.815 mol/L of glycerol. The white precipitate was obtained by adding a specific amount of 2.5 M NH_3_ (aq.) into the mixture. The precipitate was rinsed with deionized water and dried at 80 °C for 24 h to get TiO_2_ powder. The colored TiO_2_ NPs were produced upon subsequent calcination of the powder at 300 °C for 1 h. The process yielded 94% of the product.

Herein, glycerol can play a role as both reducing agent and a co-solvent [27]. Earlier reports suggested that glycerol can induce a relatively fast reduction of metal salts, and glycerol seems to be the solvent of choice for material synthesis, especially those involving metallic precursors [33]. Furthermore, the degree of reduction can be simply controlled by controlling the concentration of glycerol and the byproducts of glycerol can be easily removed by washing. In contrast, Fang et al. [15] prepared TiO_2−*x*_ by NaBH_4_ reduction at high temperature. However, the large amount of boron hydroxide produced during the reduction process and adsorbed on the TiO_2_ surface was difficult to remove. In the current work, the initial reaction was carried out in a cooling ice bath and the sample was calcined at low temperature. The active hydrogen was produced in-situ as opposed to previous studies [34], which was helpful in maintaining the original surface morphology of TiO_2_ NPs. Various colored (off-white, light grey, dark grey, and black) TiO_2_ NPs were prepared by varying the concentration of glycerol. Numerous analytical techniques were employed for characterization to identify and explain the effect of glycerol on properties of TiO_2_, including color, crystalline phase, light absorption, bandgap, surface morphology, and surface area.

### 3.2. Characterization of the Material

#### 3.2.1. Structural, Physical, and Optical Properties

Figure 2 shows the Raman spectra of the synthesized colored TiO_2_ NPs. Raman active modes corresponding to anatase crystalline structure with frequencies located at 167, 399, 515, and 640 cm^−1^ were detected for all the prepared TiO_2_ NPs, consistent with the previous study [35]. Raman scattering of the anatase TiO_2_ usually exhibits six vibrational modes composed of 3E_g_ + 2B_g_ + A1_g_ [23]. It should be noted that only four Raman active modes with corresponding peaks—167 cm^−1^ (E_g_ v5), 399 cm^−1^ (B1_g_, v3), 515 cm^−1^ (A1_g_ + B1_g_, v1 + v2), and 640 cm^−1^ (E_g_ v4)—were observed for colored TiO_2_ NPs either prepared in water or aqueous glycerol solution. No additional vibrational modes of rutile or brookite or shifting of the scattering peaks to higher wavenumbers were observed with the addition of glycerol. However, the Raman scattering peaks became weaker in intensity with increasing glycerol concentration to 1.163, 3.834, and 5.815 mol/L. It has been reported elsewhere that colored TiO_2_ exhibits weakening of the scattering peaks compared to white TiO_2_ [13,36]. The weakening and broadening of the scattering peaks is an indication of the structural disorder or the presence of reduced species like Ti^3+^. The existence of Ti^3+^ species in the colored TiO_2_ NPs prepared in 1.163, 3.834, and 5.815 mol/L aqueous glycerol solution and calcined at 300 °C was confirmed by X-ray photoelectron spectroscopy (XPS) analysis. Comparing the Raman spectra with those of black anatase titania developed by Myung et al., different features were found for our prepared colored TiO_2_ NPs. The key difference was the shifting of scattering peaks in their prepared black titania to higher wavenumbers. We do not know the exact reason for the discrepancy between our findings and those of Myung and co-workers, but it might reflect the difference in synthetic methods, especially by the addition of glycerol to obtain colored TiO_2_ NPs, which differ from black TiO_2_ prepared by the addition of urea and HF [37].

As illustrated in Figure 3a, the color of TiO_2_ changed from an off-white to light grey, dark grey, and black with increasing glycerol concentration. The sample prepared without glycerol (T1) showed an off-white color. The color of T2, T3, and T4 turned light grey, dark grey, and black, with increasing glycerol concentration to 1.163, 3.384, and 5.815 mol/L, respectively. The results indicate the effect of glycerol on the color of TiO_2_ NPs. Previously, the color of the material was correlated with the absorption of light. The color of the material appears completely black if it absorbs 100% light across the entire visible region. The equal absorption of only a certain percentage across the visible range will give a partly black or grey color [17].

The corresponding DR UV-Vis spectra are shown in Figure 3b. The absorption spectra show a narrow primary absorption edge starting at ~400 nm for T1. No significant visible light absorption for T1 was witnessed due to its wide bandgap (3.16 eV). The optical absorption edges were extended into the visible spectral range with increasing glycerol concentration. Sample T2 shows the primary absorption edge at above 400 nm, with a secondary absorption edge at around 600 nm. The other samples, T3 and T4, show primary absorption edges at ~500 nm, while their secondary absorption edges were extended well into the visible region (~650 nm). A gradual increase in the intensity of the absorption bands could be witnessed with an increase in glycerol concentration. Our findings are in agreement with Myung et al., who found that the color of white TiO_2_ turned completely black with increasing hydrofluoric concentration [37]. Our results demonstrate that glycerol mediated synthesis of TiO_2_ not only changed its color but also significantly improved the visible light harvesting property ranging from 400 to 800 nm. The optical absorption was clearly extended into the visible, and even the infrared, region of the light spectrum upon increasing glycerol concentration. The absorption of visible light could lead to the decrease in bandgap of TiO_2_.

The bandgap energies of colored TiO_2_ NPs prepared at various glycerol concentrations were determined from reflectance (F(R)) spectra using the KM (Kubelka–Munk) function and Tauc’s relationship [38,39], as given in Equation (2):(2)$${\left[F\left(R\right)hv\right]}^{1/2}=\;K\left(hv-E_{g}\right),$$
where *K* is the constant and a characteristic of the semiconductor, *hv* represents the corresponding photon energy, and *E_g_* is optical band gap energy.

The bandgap was obtained by Tauc’s plot [*F(R)hv*]^1/2^ versus the *hv* plot, where the linear portion of the graph was extrapolated to the energy (*hv*) axis, as depicted in Appendix A. Appendix A gives a graphical representation of the decreasing trend of the bandgap with increasing glycerol concentration. The band gap of T1 was 3.16 eV. The bandgap reduced to 2.96 eV for T2 when the glycerol concentration was increased to 1.163 mol/L. The bandgap was further reduced to 2.50 eV and 2.29 eV for T3 and T4 with increasing glycerol concentration to 3.834 mol/L and 5.815 mol/L, respectively (Table 2). There is therefore evidence that bandgap was significantly reduced in the presence of glycerol during the synthesis of colored TiO_2_ NPs. Consequently, the photocatalytic performance can be expected to enhance due to narrow bandgap and improved light absorption in the visible region. We believe that the reduction in bandgap could be due to the presence of Ti^3+^ species in colored TiO_2_ NPs prepared in aqueous glycerol solution. In fact, XPS analysis suggests the presence of Ti^3+^ ions in our prepared colored TiO_2_ NPs. Similar results of bandgap narrowing in the presence of Ti^3+^ ions were previously reported for black TiO_2_ [10,22,40]. Furthermore, the valance band and conduction band positions of the colored TiO_2_ were determined from XPS results using Equations (3) and (4), respectively [41]:(3)$$E_{VB}=X-\;E_{c}+0.5\;E_{G},$$
(4)$$E_{CB}=\;E_{VB}\;-\;E_{G},$$
where X = 5.81 eV and represents the absolute electronegativity (EN) of TiO_2_, $E_{c}$ denotes the energy of free electrons (4.5 eV against NHE), and $E_{G}$ denotes the band gap energy of the material calculated through Equation (2).

The valance band (VB) locations of all the TiO_2_ NPs are shown in Figure 3c. The VB and conduction band (CB) of T1 were located around 2.60 and −0.56 eV, respectively. A shift of the VB and CB was observed with increasing glycerol concentration. Compared to T1, the VB and CB positions of T2, T3, and T4 were shifted to 2.48 and −0.48 eV, 1.76 and −0.74 eV, 1.94 eV and −0.39 eV, respectively. The band structure tailoring through varying glycerol concentration caused a redshift in the optical response and created a midgap state in colored titania NPs. The electron in colored TiO_2_ can be easily excited from the VB to the CB and better charge separation can be achieved by efficient visible light absorption [42].

#### 3.2.2. Textural Properties and Morphology

Figure 4 shows the N_2_ adsorption–desorption isotherms of the synthesized colored TiO_2_ NPs. The textural analysis indicates that all the colored TiO_2_ NPs were essentially mesoporous materials presenting type IV isotherms and the H_2_ type hysteresis loop. The hysteresis loop shows a steeper desorption branch at a high relative pressure, which is typical of mesoporous material with small pore size distribution [43]. The adsorption isotherms of the as-prepared colored TiO_2_ NPs exhibited a rise in the range of 0.4–0.6 (P/P_o_), which indicates the capillary condensation ensuing monolayer formation within mesopores [44]. As can be seen in Figure 4d, T4 showed a hysteresis loop with a less sloping adsorption branch and a steeper desorption branch at a high P/P_o_ range (0.4–0.8). This can be due to the distribution of different sized pores with an equal entrance diameter [44]. In addition, T4 showed an inflection of N_2_ adsorbed volume at P/P_o_ = 0.4, indicating the presence of a well-developed mesopore. The specific surface area, average pore size, and pore volume of all the colored TiO_2_ NPs are listed in Table 2.

Appendix A illustrates the comparison of pore size distribution of the colored TiO_2_ NPs. The pore size distributions were estimated from the desorption data, employing the BJH method [43]. We can observe from Appendix A that the pore size plot shows a smaller average pore size of the TiO_2_ NPs prepared without glycerol (T1) centered at 4.35 nm and its inflection point was located at P/P_o_ = 0.4 (Figure 4a). In T2, with the addition of glycerol (1.163 mol/L), the inflection point shifted to a higher relative pressure range of 0.5–0.6 (Figure 4b), and the average pore size was less than 4.0 nm (Appendix A).

From inspection of Table 2, it emerges that the BET specific surface area decreased from 154.52 m^2^/g to 99.88 m^2^/g, along with the shrinkage of pore volume from 0.198 to 0.161 cm^3^/g (Table 2). As shown in Figure 4c,d, T3 and T4 revealed the same inflection points (P/P_o_ = 0.4). However, their specific surface area was reduced to 40.35 m^2^/g and 12.01 m^2^/g, along with a decrease in pore size to 3.06 nm and 4.47 nm (Appendix A) when the glycerol concentration was increased to 3.834 mol/L and 5.815 mol/L, respectively. The pore volume of T3 and T4 also showed a decrease to 0.030 and 0.014 cm^3^/g, respectively, with an increase in glycerol concentration. Despite the discrepancy among different TiO_2_ samples in the present work, the colored TiO_2_ NPs showed larger surface areas, except for T3 and T4 compared to the previously reported commercial TiO_2_ photocatalyst (Degussa P25, 50 m^2^/g). The decrease in surface area can be attributed to increasing particle size, as was inferred from particle size range (FESEM results). In contrast to the effect of glycerol on the particle size, the specific surface area, pore size, and pore volume showed a declining trend with increasing glycerol concentration. The pore size distributions of all the prepared colored TiO_2_ NPs were relatively narrow, indicating the good uniformity of the pores.

The samples were further analyzed with FESEM to investigate the effect of glycerol on the surface morphology of the as-synthesized colored TiO_2_ NPs. The average particle size range of the colored TiO_2_ NPs estimated from FESEM images are given in Table 2. FESEM micrographs of the TiO_2_ NPs synthesized at various glycerol concentrations are presented in Figure 5a–d. All the samples displayed nanostructure, with narrow particle size distribution ranging from 13 to 22 nm. As depicted in Figure 5a, T1 exhibited a spherical shape with particle size ranged from 13 to 17 nm. T2 hailed the same spherical shape with a little agglomeration, while its particle size increased, ranging from 16 to 22 nm, as given in Figure 5b. Compared to T1, T3 showed an almost similar average particle size range (13–18 nm) and spherical shape, indicating the insignificant effect of glycerol on particle size. Indeed, it has been shown that using glycerol as a solvent promotes smaller particle formation, which is evident from the average particle size range of T4 (11–15 nm) [45]. These results substantiated that, (1) glycerol plays an important role as a templating agent and (2) the presence of glycerol as a solvent makes the solution highly viscous. For the aforementioned reasons, the diffusion of the TiO_2_ nuclei to form larger particles and the agglomeration of the nanoparticles could be difficult or remarkably decreased.

#### 3.2.3. XPS Analysis

XPS analysis was performed to investigate the change in composition and the chemical states of Ti and O elements with the change in glycerol concentration and to correlate this with the variation in bandgap of colored TiO_2_ NPs. The XPS analysis gives an insight into the species present, change of color, and narrowing of bandgap of TiO_2_ NPs by the inclusion of glycerol as a solvent and a reductant.

Figure 6a–d shows the deconvoluted high-resolution Ti2p XPS spectra of the colored TiO_2_ NPs prepared at various glycerol concentrations and calcination at 300 °C. First, the peak positions were analyzed to know the elemental composition of the TiO_2_ surface. Figure 6a shows the Ti2p XPS spectrum of T1, which was prepared without the presence of glycerol. The Ti2p_3/2_ and Ti2p_1/2_ peaks were located at binding energies (BE) of 459.08 eV and 464.53 eV, respectively. These peak positions are the characteristic features of standard anatase phase TiO_2_. Furthermore, the calculated difference in BE of Ti2p_3/2_ and Ti2p_1/2_ (∆BE = BE Ti2p_3/2_ − Ti2p_1/2_) was equal to 5.45, which can be assigned to the typical Ti^4+^–O bonds in TiO_2_ [46]. In addition, the O1s spectrum of T1 showed only one peak centered at BE 530.95 eV (Figure 6e), which can be attributed to lattice O in TiO_2_. The calculated ∆BE between Ti2p and O1s was 71.87, which is close to that of the anatase phase (71.4 eV). The peak positions (BE values) and the difference in BE (∆BE) of (Ti2p_3/2_) − (Ti2p_1/2_) and (Ti2p_3/2_) − (O1s) of all the prepared TiO_2_ samples are provided in the Appendix A, respectively.

Compared to TiO_2_ prepared without glycerol, the Ti2p_3/2_ and Ti2p_1/2_ peaks of colored TiO_2_ NPs prepared in the presence of glycerol showed either negative or positive shift in BE. For example, the Ti2p_3/2_ and Ti2p_1/2_ peaks of T2, prepared in 1.163 mol/L aqueous glycerol solution, showed a positive shift of 0.99 eV, resulting in the peaks being centered on BE 460.07 eV and 466.30 eV, respectively (Figure 6b). This can be ascribed to the H atom in glycerol, which sustained a positive charge, pushing the Ti atom towards the neighboring O atom in the crystal lattice of colored TiO_2_ and thus plummeting the Ti–O bond length and increasing the BE of Ti2p [47]. Furthermore, an additional shoulder peak at BE of 458.80 eV (Ti2p_1/2_) was observed in T2, which is consistent with the Ti^3+^ species in the Ti_2_O_3_ lattice [48]. As can be seen in Figure 6f, the O1s spectrum of T2 exhibited two peaks positioned at BE 530.20 and 531.70, which were ascribed to lattice O in TiO_2_ and Ti_2_O_3_, respectively. This indicates the formation of TiO_2_ and some mix oxides when glycerol was used as a co-solvent and a reducing agent.

On the other hand, the Ti2p_3/2_ and Ti2p_1/2_ peaks of T3 were centered on BE of 458.5 eV and 464.65 eV, respectively, showing a negative shift of 0.13 eV (Figure 6c). The calculated ∆BE between Ti2p_3/2_ and Ti2p_1/2_ was 6.23 eV, which cannot be ascribed to the normal Ti^4+^ state in TiO_2_ and is an indication of the formation of Ti^3+^ species. Compared to T1, the O1s peak of T3 showed a negative shift of 0.75 eV in BE as can be seen from Figure 6g. Moreover, T3 exhibited a somewhat broader O1s peak at 531.20, which was located at a higher BE as compared to T1. The peak at BE 531.20 was deconvoluted with two peaks located at 530.95 and 531.70 eV, which were attributed to the lattice oxygens of TiO_2_ and Ti_2_O_3_, respectively [48]. The calculated difference in BE of (O1s) − (Ti2p_3/2_) = 72.90 eV is in reasonable agreement with that of typical Ti^3+^ containing oxides (72.9 to 73.1 eV). The oxygen in Ti_2_O_3_ lattice was reportedly located at a BE ~1.5–1.8 eV above that of O1s of anatase [49]. Similarly, as shown in Figure 6d, T4 showed a positive shift of 0.92 eV, where the Ti2p_3/2_ and Ti2p_1/2_ peaks were then centered on 460.00 and 466.02 eV, respectively. The estimated ∆BE values between Ti2p_3/2_ and Ti2p_1/2_ were 7.02 eV and an additional shoulder Ti2p_3/2_ peak at 458.32 can be attributed to Ti^3+^ species [46]. The O1s spectrum of T4 is shown in Figure 6h, where the O1s peak located 529.85 eV was deconvoluted into double peaks centered on 531.70 and 532.72, which can be assigned to lattice oxygen (O) of Ti_2_O_3_, and surface adsorbed ⁻OH, correspondingly. The results indicate the creation of Ti^3+^ species in colored TiO_2_ NPs prepared in 1.163 mol/L, 3.834 mol/L, and 5.815 mol/L aqueous glycerol solution. The shift in the position of Ti2p_3/2_ and Ti2p_1/2_ peaks suggests the impact of glycerol on the valence state of titanium element; possibly the Ti^4+^ was reduced by glycerol, which is indicated by the presence of additional peaks analogous to Ti^3+^. The previous reports on black titania suggest that additional peaks will appear in the Ti2p XPS spectrum when the valence state of Ti changes (Ti^4+^ is reduced Ti^3+^) or the Ti2p peaks will exhibit a negative shift in binding energy [50].

Next, the variation in the stoichiometry was calculated from the change in the relative peak area of Ti2p to determine how much of the Ti^4+^ and Ti^3+^ was present on the surface of the prepared colored TiO_2_. T1 did not show any peak corresponding to Ti^3+^ species. Therefore, the change in stoichiometry was calculated only for T2, T3, and T4. The peak area of Ti^3+^ increases and the Ti^4+^ peak area decreases with increasing the concentration of glycerol. The peak area of Ti^3+^ increased by ~11.65% and that of Ti^4+^ reduced by ~33% for T2, as shown in Figure 6b. As can be seen from Figure 6c, compared to T2, the Ti^3+^ peak area of T3 increased by ~33.85% while, that of T^4+^ decreased by ~86.15%. The increase in the Ti^3+^ peak area indicates the formation of Ti_2_O_3_, while the decrease in the area of Ti^4+^ peak suggests the reduction of TiO_2_ (TiO_2−*x*_). Increasing the concentration of glycerol increased the amount of Ti^3+^ ions. These results are in agreement with the color change of the TiO_2_—as the amount of Ti^3+^ increased on the surface the color became darker, as shown by photographs of the colored TiO_2_ in Figure 3a. Previous reports suggest that Ti^3+^ and/or SOVs turn white TiO_2_ into yellow, blue, and black material [51]. The decrease in the peak area corresponding to Ti^4+^ could be due to the interaction of Ti^4+^ with electrons coming from H in glycerol. Now, as observed from DR UV-Vis analysis, the bandgap reduced from 3.16 (T1) to 2.96 eV, 2.50 eV, and 2.29 eV for T2, T3, and T4, respectively. This can be correlated with the reduction in TiO_2_ by glycerol and the presence of Ti^3+^ species. As we know that the samples were prepared in the presence of glycerol, the possible redshift/reduction in the bandgap energy as observed in DR UV-Vis spectra was the introduction of donor states (here Ti^3+^ ions) in the energy gap. Liu et al. reported that the SOVs or Ti^3+^ induced occupied gap states that improve the visible light harvesting by black hydrogenated TiO_2_ produced by the reduction of NaBH_4_ [52].

Here again, the variation in the stoichiometry of oxygen was calculated from the change in the area of relative peaks. Compared to T1, a shoulder peak appeared at BE 531.7 eV in the O1s spectrum of T2, as given in Figure 6e. As shown in Figure 6f, the area of this peak increased by ~43.69% with increasing glycerol concentration from 1.163 to 3.834 mol/L. This finding is analogous to the Ti2p spectrum, where the Ti^3+^ area increased by ~33.85%, suggesting the formation of Ti_2_O_3_. As the glycerol concentration was increased to 5.815 mol/L, the area of the peak located at BE 531.7 and corresponding to oxygen in Ti_2_O_3_ was increased by ~25.32%. The XPS analysis indicated the presence of Ti^+3^, and the formation of Ti_2_O_3_. In contrast to previous work [22], no surface oxygen vacancies were created in glycerol-mediated synthesized colored TiO_2_ NPs.

### 3.3. Photocatalytic Activity

Palm oil mill effluent (POME) is the major waste effluent produced by the oil palm agroindustry during the extraction of palm oil. Conventional biological methods viz anaerobic digestion and anaerobic ponding system, commonly adopted in Southeast Asia for POME treatment, are virtually ineffective in the complete degradation of phenolic compounds. Thus, even after the treatment, phenolic compounds and color persist in treated POME, and it has been reported to contain high levels of phenolic compounds ranging from 33 to 630 ppm as GAE [30,53]. In addition, other constituents, such as the tannins and lignin [54], phenolic compounds are the major component responsible for the brown color of treated POME [55,56]. The photocatalytic performance of the colored TiO_2_ NPs synthesized at various glycerol concentration and calcined at 300 °C was evaluated for the degradation of phenolic compounds and color removal from treated POME under visible light irradiation.

The TiO_2_ samples were suspended in the treated POME solution and stirred for 30 min without light to achieve the adsorption–desorption equilibrium of phenolic compounds on the photocatalyst surface. The normalized concentration (C/C_o_) of phenolic compounds and color versus irradiation time is shown in Figure 7a,b, respectively. During controlled experiments performed in the presence of light without any photocatalyst, minor photolysis was observed which was not more than 2%. A noticeable improvement in phenolic compounds degradation and color removal was witnessed when the colored TiO_2_ NP were suspended in treated POME solution, suggesting the appropriate photocatalytic activity of the nanoparticles. It can be seen from Figure 7a that T2 prepared in 1.163 mol/L aqueous glycerol solution showed distinctly better photocatalytic performance and the degradation efficiency of phenolic compounds reached 48.17% after 180 min of visible light illumination. The color removal efficiency rose to 62.81% by T2 prepared in 1.163 mol/L aqueous glycerol solution, as shown in Figure 7b. Tan et al. also investigated the photocatalytic treatment for the removal of color from treated POME using Degussa P-25 photocatalyst [57]. However, they could only remove 50% of the color from treated POME in 240 min of UVB irradiation. It is important to note here that almost 62.81% of the color was removed from treated POME, which is an indication that no excessive intermediates or by-products were generated during the degradation process. A common characteristic of the presence of by-products of phenolic compound degradation is a yellow coloration of the solution [58]. However, in the present study, the final solution was colorless and transparent, which indicates the complete degradation of phenolic compounds. In comparison to other studies, the improved photocatalytic performance is most likely because of the presence of Ti^+3^ species [59] and efficient light absorption in the visible region by T2. Such improved photocatalytic performance of grey TiO_2_ nanowires produced via Al reduction for the decomposition of organic dyes was reported by Yin and co-workers [23]. The enhanced photocatalytic performance of colored TiO_2_ has previously been associated with Ti^3+^ ions’ [60] narrower bandgap and better charge separation [61]. The better photocatalytic performance of the colored TiO_2_ prepared in 1.163 mol/L aqueous glycerol solution can also be attributed to the formation of Ti_2_O_3_, which is a stoichiometric compound of the lower valance state of (Ti^3+^), and a more robust atomic structure compared to TiO_2−*x*_, which is oxygen deficient. Ti_2_O_3_ is important for maintaining and stabilizing Ti^3+^ species on the surface, which is key to the improved photocatalytic performance of black titania [62]. T3 showed only 32.41% phenolic compounds degradation and 53.66% of color removal from treated POME. The phenolic compound degradation efficiency further dropped to 27.68% for T4. The results suggest that as the glycerol concentration increased from 1.163 mol/L to 5.815 mol/L, the photocatalytic performance of the colored TiO_2_ NPs decreased. This can be due to the increasing amount of Ti^3+^ species, which is obvious from XPS analysis where the amount of Ti^3+^ species increased with increasing glycerol concentration. Several scientific reports suggest that inducing more additional localized electronic states such as Ti^3+^ below the conduction band minimum might be detrimental to achieving improved photocatalytic performance [63,64]. The results suggest that visible light photoactive colored titania nanoparticles with the appropriate amount of Ti^3+^ species, high stability and surface area for the degradation of phenolic compounds and color removal from treated POME can be produced starting from a lower concentration of glycerol.

The stability of the photocatalyst is an important consideration for industrial-scale application in wastewater treatment. To demonstrate the stability of the synthesized colored TiO_2_ photocatalyst, T2 was reused for the recycling tests for three consecutive runs due to its higher photocatalytic performance. After each run, the photocatalyst was retained by centrifugation and washed several times with deionized water. The results of the recyclability tests are shown in Figure 7c. Only a slight decrease, not more than 3%, can be observed after the three consecutive runs, and the photocatalytic activity was well retained. The results indicate that glycerol-mediated synthesized TiO_2_ NPs are very stable under the employed reaction conditions.

In the current study, the photocatalytic reactions were performed on actual wastewater matrix (treated POME), which is a complex mixture of several phenolic compounds, including caffeic acid (3,4 dihydroxycinnamic acid, C_9_H_8_O_4_), ferulic acid (dihydroxycinnamic acid, C_10_H_10_O_4_), gallic acid (3,4,5 trigydroxybenzoic acid, C_7_H_6_O_5_), Phenol (C_6_H_5_OH), Catechol (1,2-dihydroxybenzene, C_6_H_6_O_2_), 3-Methylcatechol (C_7_H_8_O_2_), 4-Hydroxybenzoic acid, p-Coumaric acid, Protocatechuic acid (C_9_H_8_O_3_), and p-hydroxybenzoic acid. Therefore, it was not possible to carry out a mechanistic investigation to monitor the evolution of intermediates formation during the degradation process and propose a suitable degradation mechanism. However, previous studies which used gallic acid as a single model pollutant proposed suitable degradation mechanisms [65,66]. For instance, Luna and co-workers [67], investigated photocatalytic degradation of gallic acid in an aqueous solution and proposed the plausible mechanism of degradation, as shown in Figure 8. Their results showed that four intermediates, maleic, fumaric, oxalic, and formic acid, were identified during the photocatalytic degradation of gallic acid.

## 4. Conclusions

A series of colored, anatase TiO_2_ nanoparticles were synthesized via a simple chemical precipitation method in the presence of glycerol. Interestingly, the use of glycerol, which acted as co-solvent and a reductant, provided a green route to synthesize colored titania with a narrow bandgap and significantly superior photocatalytic performance. The enhanced photocatalytic performance was confirmed by the efficient degradation of phenolic compounds (48.17%) and color removal (62.18%) from treated POME matrix under visible light irradiation for 180 min. The enhanced visible light absorption capability and photocatalytic performance can be ascribed to the introduction of Ti^3+^ species as Ti_2_O_3_. Based on the results, it was found that the concentration of glycerol is a determining factor for the extent of reduction, change of color, variation in specific surface area, bandgap, and the formation of Ti^3+^ species.

## Figures and Tables

**Figure 1 nanomaterials-09-01586-f001:**
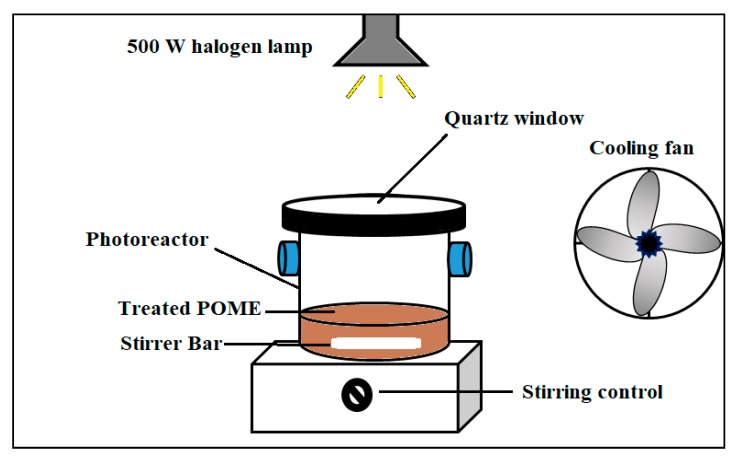
Schematic sketch of the experimental setup for photocatalytic reactions.

**Figure 2 nanomaterials-09-01586-f002:**
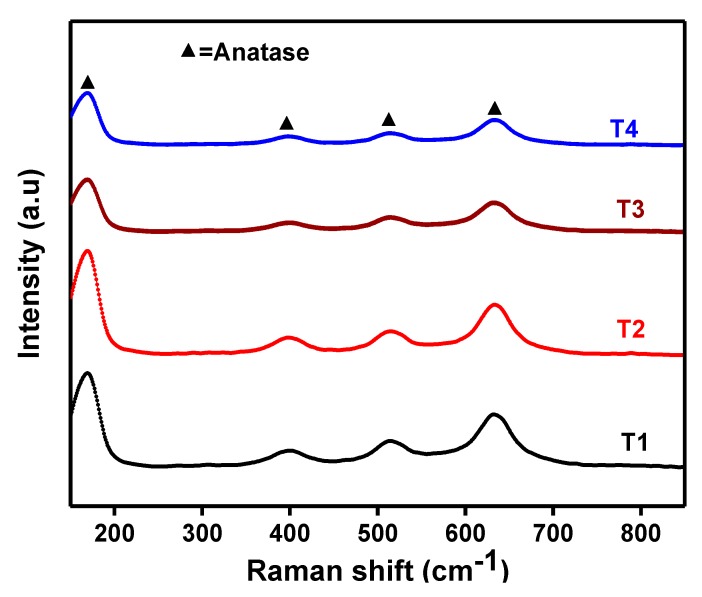
Raman spectra of the colored TiO_2_ nanoparticles (NPs) prepared at various glycerol concentrations and calcined at 300 °C.

**Figure 3 nanomaterials-09-01586-f003:**
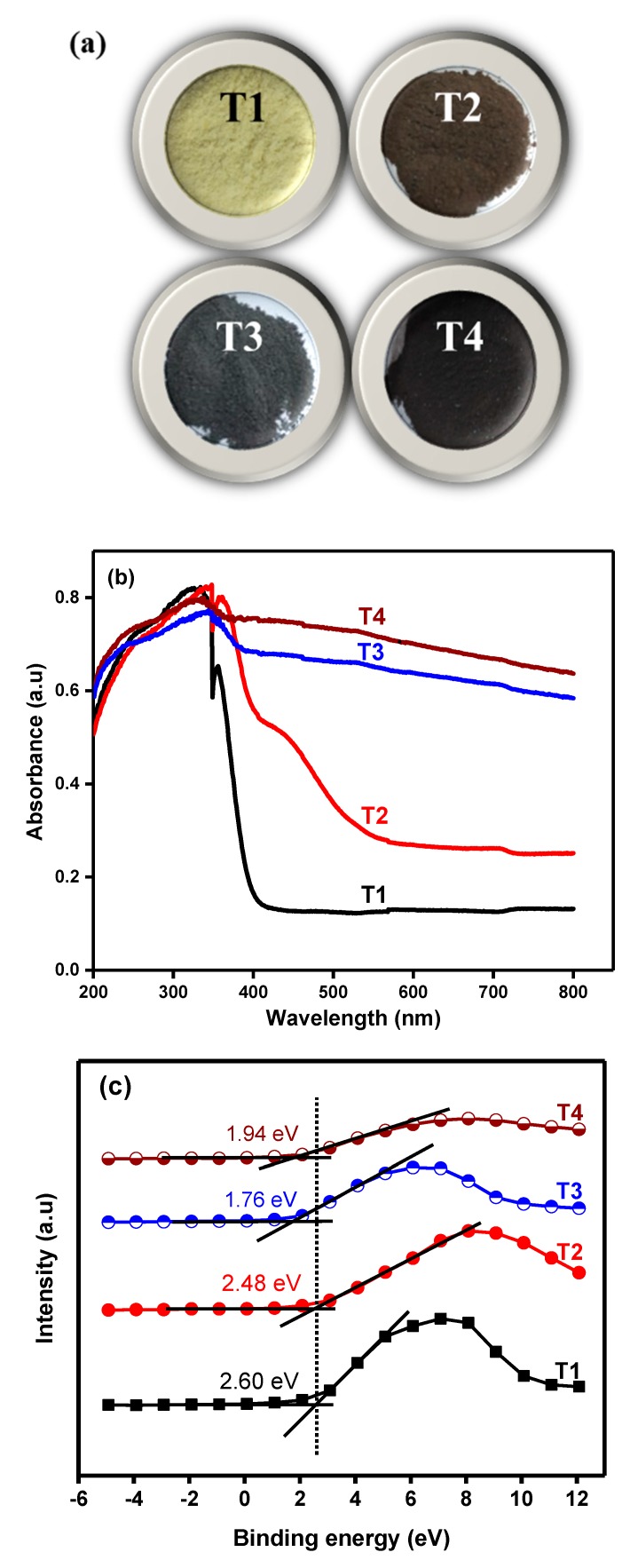
(**a**) Photographs, (**b**) DR UV-Vis absorption spectra, and (**c**) valence band positions of colored TiO_2_ NPs prepared at various glycerol concentrations and calcined at 300 °C.

**Figure 4 nanomaterials-09-01586-f004:**
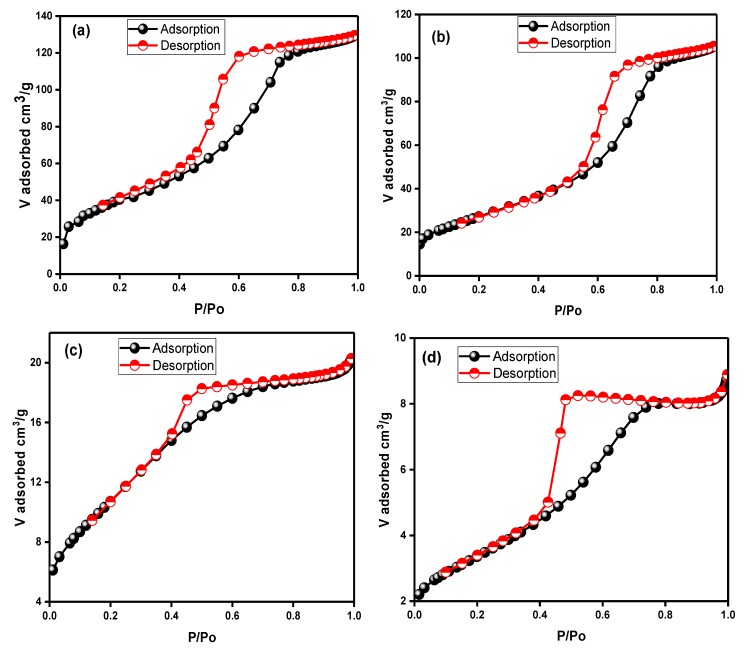
N_2_ adsorption–desorption isotherms of colored TiO_2_ prepared at various glycerol concentrations and calcined at 300 °C (**a**) T1, (**b**) T2, (**c**) T3, and (**d**) T4.

**Figure 5 nanomaterials-09-01586-f005:**
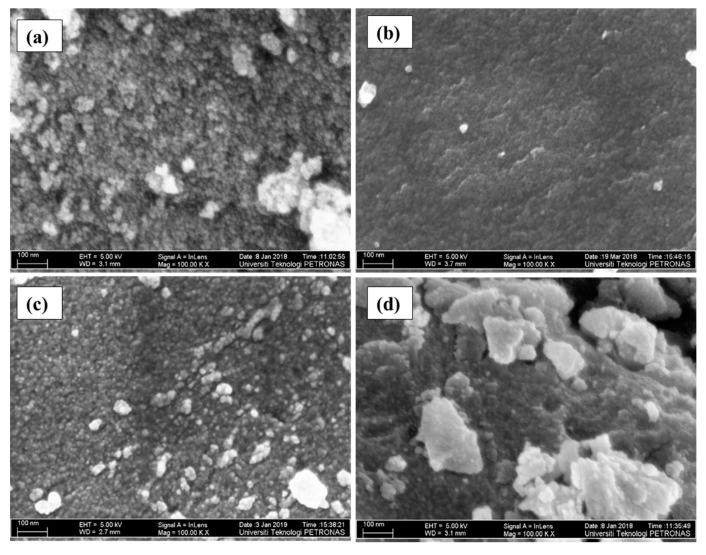
FESEM photographs of the colored TiO_2_ prepared at various glycerol concentrations and calcined at 300 °C; (**a**) T1, (**b**) T2, (**c**) T3, and (**d**) T4.

**Figure 6 nanomaterials-09-01586-f006:**
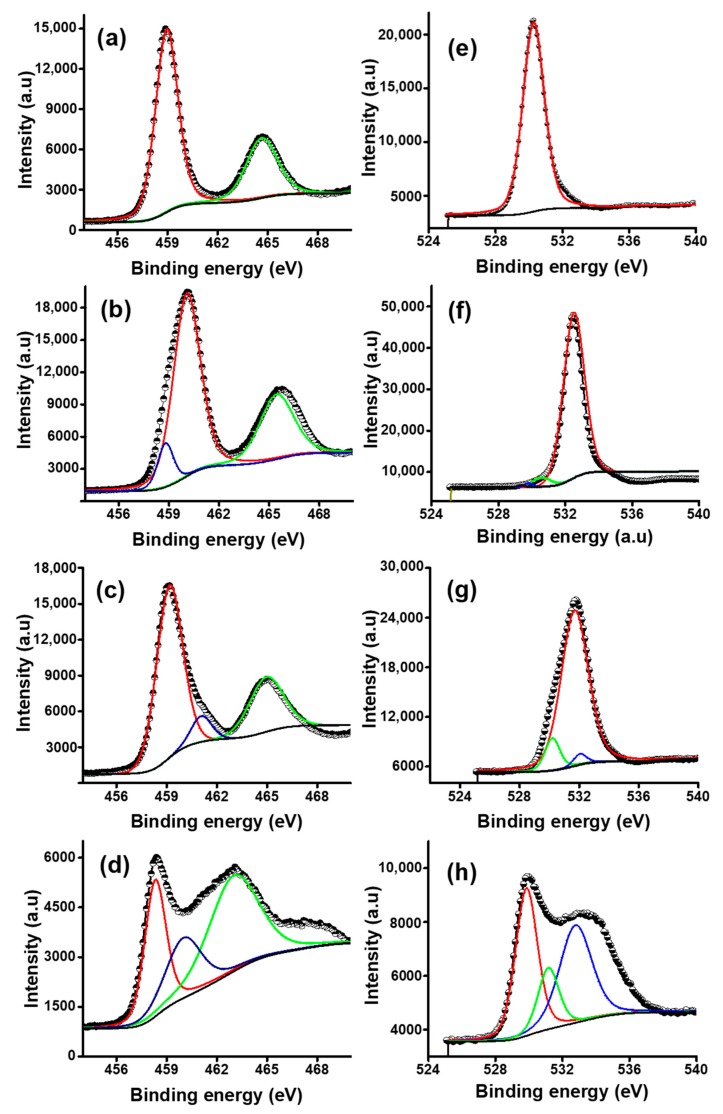
High resolution XPS spectra of Ti_2_p (**a**–**d**) and O1s (**e**–**h**) of TiO_2_ prepared at various glycerol concentration and calcined at 300 °C (**a**) T1 (**b**) T2, (**c**) T3, (**d**) T4, and O1s for (**e**) T1, (**f**) T2, (**g**) T3, and (**h**) T4.

**Figure 7 nanomaterials-09-01586-f007:**
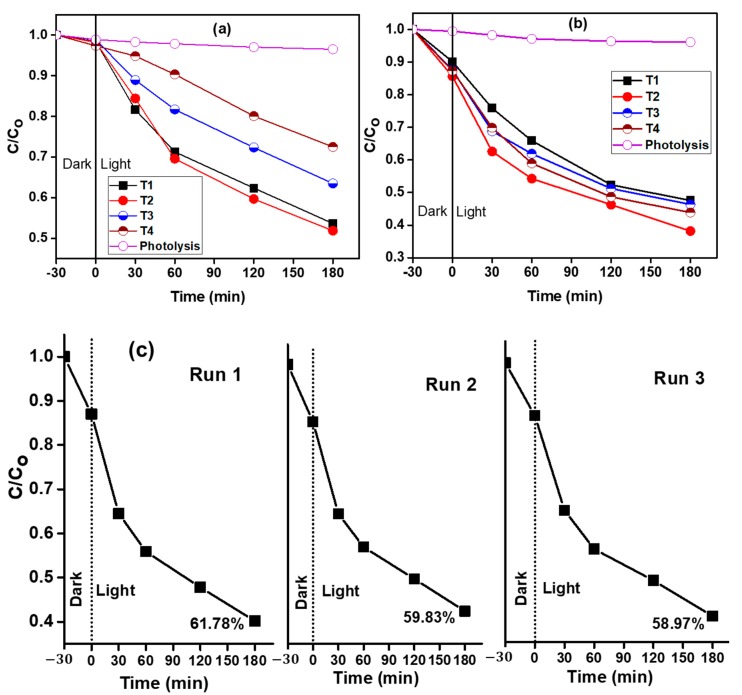
Photocatalytic efficiency: (**a**) phenolic compounds degradation, (**b**) color removal of the colored TiO_2_, and (**c**) recycling results of T2 in decolorization of treated palm oil mill effluent (POME).

**Figure 8 nanomaterials-09-01586-f008:**
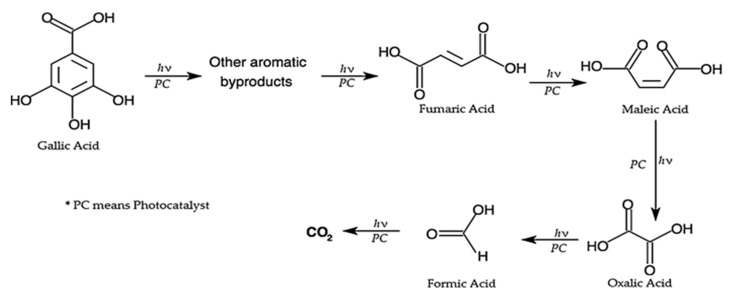
Proposed mechanism for the photocatalytic degradation of gallic acid reproduced from [67], with permission from Elsevier, 2016.

**Table 1 nanomaterials-09-01586-t001:** Concentration of aqueous glycerol expressed as glycerol:water volume ratio and molarity for the synthesis of colored TiO_2_ nanoparticles.

Sample Label	Glycerol:H_2_O (*v*/*v*)	Glycerol Concentration (mol/L)
T1	0:1	0.0
T2	1:9	1.163
T3	1:2	3.834
T4	1:1	5.815

**Table 2 nanomaterials-09-01586-t002:** Properties of the colored TiO_2_ nanoparticles synthesized at various glycerol concentrations and calcined at 300 °C.

Sample	Band Gap/eV (±0.1)	Specific Surface Area/m^2^·g^−1^ (±5%)	Pore Volume/cm^3^·g^−1^ (±0.01)	Average Pore Size/nm (±0.01)	Particle Size Range (nm)
T1	3.16	154.51	0.198	4.35	13–17
T2	2.96	99.88	0.161	6.49	16–22
T3	2.50	40.35	0.030	3.06	13–18
T4	2.29	12.01	0.014	4.47	11–15

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
