# Peer review of "Glycerol-Mediated Facile Synthesis of Colored Titania Nanoparticles for Visible Light Photodegradation of Phenolic Compounds"

_nanomaterials, 2019, doi:10.3390/nano9111586_

Round 1

Reviewer 1 Report

The current manuscript deals with the synthesis of colored titania nanoparticles for enhanced photocatalytic performance. This manuscript fits well to the Nanomaterials. However, only some aspects have novelty and the title of the manuscript does not fit very well with the content. Below the authors will find some comments:

It is very important that the authors define a research task or question for their study. They should mention them in the last part of section 1. Introduction.

Overall, the manuscript is too long. The results of different studies concerning the structural, physical and optical properties have been published from different authors. Therefore, this section can be shortening.

The authors should focus on the title – enhanced photocatalytic performance (e.g., analyse of the UV-light effects); Are all methods used in this study necessary to fulfil the research task? At the moment, the study of the photocatalytic activity is very short described.

Reviewer 2 Report

Nanomaterials-621471

Title: Glycerol-Mediated Facile Synthesis of Colored Titania Nanoparticles for Enhanced Photocatalytic Performance

This paper deals with a study on the fabrication of colored titania nanoparticles (NPs) by a new glycerol-mediated safe and facile method. Also, authors successfully optimized colored titania nanoparticles application for efficient and best photocatalytic performance under natural solar light. The study brought out some impressive results in the photocatalyst research interest and benefited for science and society.

Nevertheless, the manuscript also suffers from some problems that question the relevance of gained results. Primary interrogations are as mentioned below. Overall after working on interrogations,I suggest this paper to be accepted suitable for publication in Nanomaterials

TITLE “Glycerol-Mediated Facile Synthesis of Colored Titania Nanoparticles for Enhanced Photocatalytic Performance.” The title fits well with the article, however its better if the title emphasizes the applicability developed colored catalyst under visible/solar light

ABSTRACT looks clear explains the purpose of the research, the principal results, and significant conclusions.

INTRODUCTION

The introduction is pleasant to read and provides an accurate view of the current state of the art. Some minor writing errors must be corrected prior to publication.

EXPERIMENTAL

0.100 mole of check this

Synthesis of colored TiO2 nanoparticles

The author needs to revise this section it’s better to mention exact volumes and amount utilized for the synthesis and provide the yield of the product.

Check the caption for Table 1. TiO2 samples synthesized at various glycerol concentrations and calcined at 300 ºC.

RESULTS AND DISCUSSION

Photocatalytic activity

What are the degradation components of phenol authors that need to confirm by advanced techniques? Also, authors must provide the detail information related to color giving component in the contaminated color solutions and adsorption wavelengths. In Figure 7. authors must update figures with a control experiment in the absence of catalyst. Especially for colored component degradation experiment The author need to confirm the stability of the photocatalyst residue after the treatment with phenol and colored contaminant. Authors must provide the experiments for the reusability of the catalyst.

Round 2

Reviewer 1 Report

The authors reworked the mansucript according the comments of the reviewers. The red thread of the manuscript can be seen.

Reviewer 2 Report

This is an improved version of the manuscript. The authors nicely worked on all raised questions.

But still, manuscript lacking with a mechanistic investigation study, authors have to justify their degradation results by suitable mechanism study. Include the most needed possible mechanism section in the manuscript. 

Authors have to confirm the stability of the photocatalyst after the reusability test. Confirm the catalyst stability by providing Before and after PXRD patterns of the catalyst.
